# Key Techniques for Somatic Embryogenesis and Plant Regeneration of *Pinus koraiensis*

**Fang Gao [1], Chunxue Peng [1], Hao Wang [1], Iraida Nikolaevna Tretyakova [2], Alexander Mikhaylovich Nosov [3,4], Hailong Shen [5,*] and Ling Yang [1,\*]**

[1] State Key Laboratory of Tree Genetics and Breeding, School of Forestry, Northeast Forestry University, Harbin 150040, China; fanggao@nefu.edu.cn (F.G.); chunxue@nefu.edu.cn (C.P.); 15636830205@163.com (H.W.)

[2] Laboratory of Forest Genetics and Breeding, Institution of the Russian Academy of Sciences V.N., Sukachev Institute of Forest Siberian Branch of RAS, 660036 Krasnoyarsk, Russia; culture@ksc.krasn.ru

[3] Department of Cell Biology and Institute of Plant Physiology K.A., Timiryazev Russian Academy of Sciences, 127276 Moscow, Russia; al_nosov@mail.ru

[4] Department of Plant Physiology, Biological Faculty, Lomonosov Moscow State University, 119991 Moscow, Russia

[5] State Forestry and Grassland Administration Engineering Technology Research Center of Korean Pine, Harbin 150040, China

\* Correspondence: shenhl-cf@nefu.edu.cn (H.S.); yangl-cf@nefu.edu.cn (L.Y.)

**Abstract:** Korean pine is the dominant species of Korean pine forests. It is an economically valuable species that yields oil, high-quality timber and nuts, and it offers great prospects for further development. Complete regenerated plants of Korean pine were obtained via somatic embryogenesis using megagametophytes as the explant. The seeds of 27 families of Korean pine were collected to induce embryogenic lines. We compared the effects of explant collection time, family and medium components (concentrations of sucrose, plant growth regulators and acid-hydrolyzed casein) on embryogenic lines induction. The effects of plant growth regulators and L-glutamine contents on the proliferation and maturation of embryogenic cell lines were studied, and the germinating ability of different cell lines was evaluated. The embryogenic lines induction percentage of Korean pine reached 33.33%. When 4.52 μmol·L$^{-1}$ 2,4-D and 2.2 μmol·L$^{-1}$ 6-BA were added to the medium of embryogenic lines proliferation, the ability of embryo maturation was the best (cell line 001#-100 was 135.71·g$^{-1}$ fresh weight). Adding 1–1.5g L$^{-1}$ L-glutamine to the proliferation medium can improve the ability of embryo maturation (cell line 001#-100 was 165.63·g$^{-1}$ fresh weight). The germination percentage of the three cell lines tested was significant, and the highest was 66%. We report on successful regeneration and cryopreservation methods for somatic embryos of Korean pine. This technology could be used to propagate the excellent germplasm resources of Korean pine and to establish multi-varietal forestry.

**Keywords:** Korean pine; embryogenic lines; somatic embryos; regenerated plant; megagametophytes; cryopreservation

---

## 1. Introduction

Somatic embryogenesis is the formation of embryo-like structures from somatic cells without gametic fusion [1,2]. Somatic embryos (SEs) are not only the best receptor system for genetic transformation [3,4], but are also invaluable for the large-scale propagation of excellent germplasm resources, and have many potential applications [5–7]. To date, somatic embryogenesis has been achieved for nearly 30 pine species. This method is used in production for some species, and has had remarkable economic benefits [8].

Korean pine (*Pinus koraiensis* Sieb. Et Zucc.) is a species in the Pinaceae that is mainly distributed in northeast China, the far east of Russia, the Korean Peninsula and Japan [9]. Korean pine is an economically valuable species that yields oil, high-quality timber and nuts, and it has great prospects for further development [10]. In recent decades, the populations of natural broad-leaved Korean pine mixed forests have been sharply reduced [11,12], and the species has been listed as nationally endangered in China. Thus, the protection of *P. koraiensis* has become urgent [12]. The development of powerful clonal propagation methods, such as somatic embryogenesis, has potentially numerous application advantages over conventional rooted cuttings [13]. Somatic embryogenesis is considered the main way to achieve true rejuvenation in vegetative propagules, because SEs develop both embryonic apical and root meristems [14].

Somatic embryogenesis comprises the coordinated execution of four steps: embryogenic lines (EC) induction, proliferation, maturation and plant regeneration [15]. One of the main factors affecting SEs induction in pines is the developmental stage of the zygotic embryo [16]. Immature embryos are the most successful explant for the somatic embryogenesis of pine species, and the induction percentage of EC from immature embryos is higher than that from mature embryos [17–19]. For most pine species, the induction percentage of EC is higher when seeds are at the cleavage polyembryony or cotyledon embryo stages of development. However, the response of different tree species is different, which needs to be determined by experiments.

Most Pinus species are recalcitrant to micropropagation and regeneration by SE [20], and Korean pine is not an exception. Bozhkov described two alternative pathways of SEs origin of *P. koraiensis* [21]. Gao induced EC and proliferated, but the induction percentage of EC was low, and SEs could not be obtained [19]. Once SEs are initiated and EC can be identified, the next challenge is to ensure the rapid proliferation of EM (embryonal mass, EM) on subculture onto fresh medium, so as to generate the amounts that are needed for various steps [22]. The quality of EC during proliferation impacts not only the yield but also the quality of the cotyledonary embryos, leading to lower germination rates [23]. At the same time, it is necessary to optimize the culture conditions of the EC proliferation stage.

In this study, the seeds from 27 families were collected. Using immature megagametophytes as the explant, we established an efficient EC induction system. The key techniques of EC proliferation and maturation were optimized. The germinating abilities of SEs of different cell lines were evaluated. These methods can be used to propagate the excellent germplasm resources of *P. koraiensis*, and to establish multi-varietal forestry.

## 2. Materials and Methods

### 2.1. EC Induction

#### 2.1.1. Selection of Explant Source

From June to July 2015, open-pollinated cones of three families (numbered 057#, 108# and 135#) were collected from the Qingshan Forest Seed Orchard (Weihe Forestry Bureau, Heilongjiang Province, China). The age of each family was 28 years, and the dates of seed collection were 23 June, 30 June, 6 July and 13 July (representing seeds in the E1, E2, E3 and E4 phases, respectively). The cones were cut from the branches and stored at 4 °C. The physiological state of explants from seeds collected at the four times was determined by observation under a microscope, and they were photographed using a Moticam 3000C camera. At the same time, 10 seeds were randomly selected, and the fresh weights of the megagametophytes were measured by peeling off the seed coat, then putting them in the oven at 108 °C at constant weights and measuring the dry weight.

On 1 July 2017, open-pollinated cones in the E1 phase were collected from 24 families, and their megagametophytes were used as explants.

### 2.1.2. Explant Sterilization Method

The cones of *P. koraiensis* were washed with detergent for about 30 min and then washed with running water for 8 h. After peeling off the seed scales, the seeds were treated with 75% (*v/v*) alcohol for 1 to 2 min on a clean bench and then washed three to five times with sterile water. The washed seeds were treated with 10% (*m/v*) sodium hypochlorite solution for 15 min, and then rinsed three to five times with sterile water. The inner and outer testae were removed, and the whole megagametophyte was sterilized by soaking in 3% (*m/v*) hydrogen peroxide for 8 min and washing three to five times with sterile water.

### 2.1.3. Screening of Medium Components

Four factors and four levels of orthogonal experiments were used. The orthogonal design is shown in Table 1. The basal culture medium was DCR [23], supplemented with 6.5 g $L^{-1}$ agar and 500 mg $L^{-1}$ L-glutamine, and the four factors were sucrose (25, 30, 35 and 40 g $L^{-1}$), 1-naphthalacetic acid (NAA) (8.06, 10.74, 13.43 and 16.11 µmol·$L^{-1}$), 2.4-dichlorophenoxyacetic acid (2.4-D) (18.10, 27.14, 36.20 and 45.24 µmol·$L^{-1}$), N6-benzyladenine (6-BA) (2.22, 4.44, 6.66 and 8.88 µmol·$L^{-1}$) and acid-hydrolyzed casein (CH) (0.3, 0.5, 0.8 and 1.0 g $L^{-1}$). The pH of the medium was then adjusted to 5.8 before autoclaving (105 Pa at 120 °C, 20 min). L-glutamine was filter-sterilized and added to the medium after autoclaving. The sterilized fertilized megagametophytes were placed into Petri dishes (Jiafeng Horticultural Products Co., Ltd., Shanghai, China), which were 90 mm diameter × 20 mm depth, and five explants per dish. The number of replicates per treatment varied in the initiation experiments. This was due to the availability of cones on sampled trees in the seed orchards. After inoculation, the cultures were kept in a culture chamber at 23 ± 2 °C in the dark. The explants were observed regularly and the EC induction percentage was calculated after 12 weeks of culture.

**Table 1.** Influencing factors of source, acid-hydrolyzed casein (CH), 1-naphthalacetic acid (NAA) and N6-benzyladenine (6-BA) on embryogenic lines (EC) induction of *Pinus koraiensis* based on orthogonal experiments design.

| Treatments | Sucrose Concentration (g $L^{-1}$) | NAA Concentration (µmol·$L^{-1}$) | 6-BA Concentration (µmol·$L^{-1}$) | CH (g $L^{-1}$) | Total Explants | Induction Percentage (%) |
|---|---|---|---|---|---|---|
| 1 | 25 | 8.06 | 2.22 | 0.3 | 125 | 20.87 ± 0.92 |
| 2 | 25 | 10.74 | 4.44 | 0.5 | 141 | 12.27 ± 1.26 |
| 3 | 25 | 13.43 | 6.66 | 0.8 | 136 | 11.66 ± 1.09 |
| 4 | 25 | 16.11 | 8.88 | 1.0 | 96 | 0.00 ± 0.00 |
| 5 | 30 | 8.06 | 4.44 | 0.8 | 97 | 14.48 ± 0.35 |
| 6 | 30 | 10.74 | 2.22 | 1.0 | 123 | 18.73 ± 1.22 |
| 7 | 30 | 13.43 | 8.88 | 0.3 | 132 | 12.17 ± 1.05 |
| 8 | 30 | 16.11 | 6.66 | 0.5 | 136 | 12.43 ± 1.05 |
| 9 | 35 | 8.06 | 6.66 | 1.0 | 138 | 21.80 ± 1.13 |
| 10 | 35 | 10.74 | 8.88 | 0.8 | 132 | 33.33 ± 0.69 |
| 11 | 35 | 13.43 | 2.22 | 0.5 | 88 | 13.65 ± 0.20 |
| 12 | 35 | 16.11 | 4.44 | 0.3 | 115 | 3.52 ± 1.34 |
| 13 | 40 | 8.06 | 8.99 | 0.5 | 157 | 11.05 ± 1.29 |
| 14 | 40 | 10.74 | 6.66 | 0.3 | 142 | 12.02 ± 1.13 |
| 15 | 40 | 13.43 | 8.88 | 1.0 | 154 | 11.63 ± 1.02 |
| 16 | 40 | 16.11 | 2.22 | 0.8 | 70 | 0.00 ± 0.00 |

Note: The values in the column Induction Percentage in the table represent the mean ± standard error.

In July 2017, the best scheme selected in 2015 was used for EC induction culture (DCR + 35 g $L^{-1}$ sucrose + 10.74 µmol·$L^{-1}$ NAA + 6.66 µmol·$L^{-1}$ 6-BA + 0.8 mg $L^{-1}$ CH + 6.5 g $L^{-1}$ agar and 500 mg $L^{-1}$ L-glutamine). Other culture methods were the same as in 2015. The sterilized fertilized megagametophytes were placed into Petri dishes, five explants per dish, and each family had 50 explants.

The EC was cytologically observed. The fresh target EC was placed on a clean microslide, stained with 0.1% safranine staining solution, and then covered with a cover glass. The cover glass was tapped gently with the flat end of a pencil to spread the plant tissue evenly. The cells were observed and photographed immediately under an optical microscope (Olympus BX51 equipped with a Moticam 3000C camera).

## 2.2. Proliferation and Maturation of the EC

### 2.2.1. Proliferation of the EC

Proliferation experiment 1: Effect of plant growth regulator (PGRs) on the proliferation of Korean Pine EC.

The EC of 001#-1, 001#-100 and 001#-34 were used as test materials. The proliferation medium was mLV [24] basic medium supplemented with 0.5 g $L^{-1}$ CH, 30 g $L^{-1}$ sucrose, 0.5 g $L^{-1}$ L-glutamine and 4 g $L^{-1}$ Gelrite (Phytagel™, Sigma-Aldrich, St. Louis, MO., USA). The hormone concentration was divided into three treatments: (1) 9.04 $\mu mol·L^{-1}$ 2,4-D + 4.4 $\mu mol·L^{-1}$ 6-BA; (2) 4.52 $\mu mol·L^{-1}$ 2,4-D + 2.2 $\mu mol·L^{-1}$ 6-BA; (3) 2.26 $\mu mol·L^{-1}$ 2,4-D + 0.44 $\mu mol·L^{-1}$ 6-BA. Embryonal mass (0.2 g) was transferred to the same composition in each subculture, each cell line with five replicates, subcultured every 2 weeks. The fresh weight of the embryogenic cell masses was measured after four subculture cycles, and the maturation ability of the embryogenic cell lines was measured under three different PGRs conditions.

Proliferation experiment 2: Effect of L-glutamine on the proliferation of Korean Pine EC.

The EC of 001#-1, 001#-100 and 001#-34 were used as test materials. The proliferation medium was mLV medium supplemented with 0.5 g $L^{-1}$ CH, 30 g $L^{-1}$ sucrose, 4.52 $\mu mol·L^{-1}$ 2,4-D, 2.2 $\mu mol·L^{-1}$ 6-BA and 4 g $L^{-1}$ Gelrite. L-glutamine concentration was divided into three treatments (0.5, 1.0 and 1.5 g $L^{-1}$). Embryonal mass (0.2 g) was transferred to the same composition in each subculture, each cell line with four replicates, and subcultured every 2 weeks. The fresh weights of the embryogenic cell masses were measured after four subculture cycles, and the maturation ability of the embryogenic cell lines was measured under three different L-glutamine conditions.

### 2.2.2. Maturation of SEs

The maturation ability of SEs was tested using the EC obtained from proliferation. The EC (100 mg) was transferred into a 50 mL centrifuge tube and the liquid proliferation medium without PGRs was added. Next, we shook the centrifuge tube violently to achieve full dispersal, and transferred the mixture to the filter paper with a pipette. A Brinell funnel was used for filtration, and then we put the filter paper with EC on the solid medium. Each cell line had between four and six replicates. The mLV medium contained 68 g $L^{-1}$ sucrose, 75.66 $\mu mol·L^{-1}$ abscisic acid (ABA), 500 g $L^{-1}$ CH, 500 g $L^{-1}$ L-glutamine and 10 g $L^{-1}$ Gelrite. The number of SEs was recorded after being cultured at 23 ± 1 °C for 3 months.

## 2.3. SEs Germination and Plant Regeneration

In total, 50 mature SEs were randomly selected from three cell lines for the germination test. The SEs needed to be desiccation treated before germination, and the desiccation treatment was completed with 6-hole cell plate (Corning-Costar 3516), in which 3 holes were filled with two layers of dry sterile filter paper. We then put the SEs on the filter paper, and the remaining three holes were filled with sterile water. Finally, the cell plates were sealed with preservative film and cultured in the dark at 4 °C for 7 days. The cotyledon embryos were placed onto the germination medium. The germination medium was mLV medium supplemented with 2 g $L^{-1}$ activated carbon, 20 g $L^{-1}$ sucrose and 4 g $L^{-1}$ Gelrite, cultured in the dark for 7 days, and then transferred to light culture (35 $\mu mol·cm^{-2}·s^{-1}$ light, 16 h light/8 h dark photoperiod, 23 ± 2 °C). The regeneration percentage of plantlets was recorded after 8 weeks of culturing.

*2.4. Transplanting and Acclimatization of Regenerated Plants*

After 16 weeks of culture, the rooted plantlets that developed from SEs were removed from the culture bottle with tweezers, the medium attached to the roots was washed off, and the plantlets were transplanted into a sterilized substrate (nutrient soil, vermiculite and perlite, at a volume ratio of 3:1:1). The plantlets were covered with plastic wrap to maintain high air humidity, and cultivated under the following conditions: 23 ± 2 °C, under a light intensity of about 35 μmol/($cm^{-2} \cdot s^{-1}$), and a 16 h light/8 h dark photoperiod. The plastic wrap was removed after 2 weeks, and the plants were watered regularly. The survival percentage of the plantlets was determined at 6 weeks after transplanting into the soil substrate.

*2.5. Microscopy Observation*

The paraffin section making method was performed as referred to by Li [25]. The cultures of different development stages were fixed in FAA fixative solution (formaldehyde: acetic acid:50% ethanol = 1:1:18). The fixed samples were dehydrated, soaked in wax and embedded in paraffin, stained with hematoxylin, then sectioned (the thickness of the section was 10 μm), and sealed with neutral balsam. The cells were observed under an optical microscope.

*2.6. Data Statistics and Analysis Methods*

The experimental data were processed using Excel 2003, and the average value and proliferation efficiency were calculated from these data. Single-factor analysis of variance (ANOVA) to evaluate the effects of PGRs, sucrose and CH was performed using SPSS 19 software (SPSS, Chicago, IL, USA). Graphs were drawn using Sigmaplot 12.0. The calculations used for the various indexes are:

$$\text{EC induction percentage}(\%) = \frac{\text{Number of explants producing EC}}{\text{Number of living explants placed}} \times 100$$

$$\text{EC proliferation efficiency}(\%) = \frac{(\text{EC quality after culture} - \text{EC quality before culture})}{\text{Fresh quality of EC before culture}} \times 100$$

$$\text{SEs per gram of EC}(g-1\,\text{FW}) = \frac{\text{Number of SEs (per petri dish)}}{\text{Fresh quality of EC before culture (mg)}} \times 1000$$

$$\text{Germination percentage }(\%) = \frac{\text{Number of SEs with new shoots}}{\text{Number of SEs placed}} \times 100$$

## 3. Results

*3.1. Induction of EC*

3.1.1. Development of Explants in Different Periods

The development state of *P. koraiensis* seeds differed among the four collection times (Figure 1), ranging from the early embryo stage (Figure 1a, Phase E1; Figure 1b, Phase E2) to the cleavage polyembryonic stage (Figure 1c, Phase E3) and prophase at the columnar embryo stage (Figure 1d, Phase E4). With the maturity of the megagametophyte, the fresh weight of the megagametophyte in three families increased first and then decreased (Figure 2a), and the dry weight increased gradually (Figure 2b).

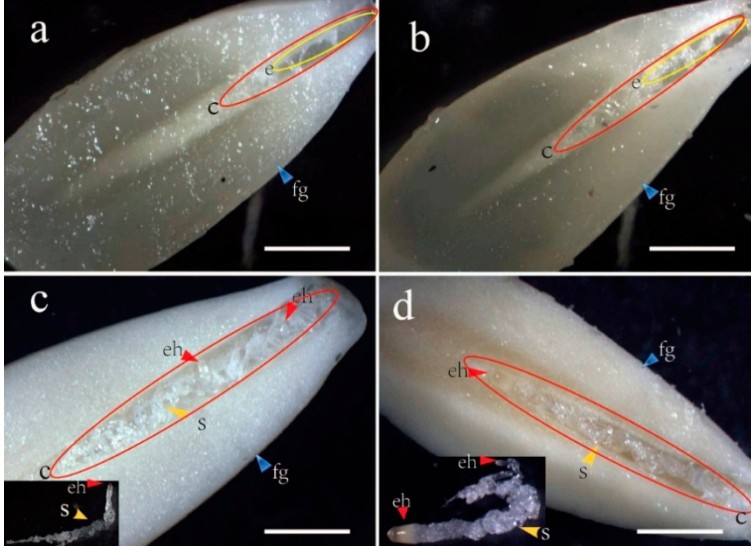

**Figure 1.** Development status of immature seeds of *Pinus koraiensis* at different collection times (2015). (**a**) Phase E1, collected on 23 June, bar = 0.2 cm; (**b**) Phase E2, collected on 30 June, bar = 0.2 cm; (**c**) Phase E3, collected on 6 July, bar = 0.2 cm; (**d**) Phase E4, collected on 13 July, bar = 0.2 cm. The red arrow points to the embryo head and is indicated by the letters 'eh', the yellow arrow points to the suspensor and is indicated by the letter s, the blue arrow points to the female gametophyte and is indicated by the letters 'fg', the yellow oval is the early embryogeny and is indicated by the letter e, and the red oval is the corrosion cavity and is indicated by the letter 'c'.

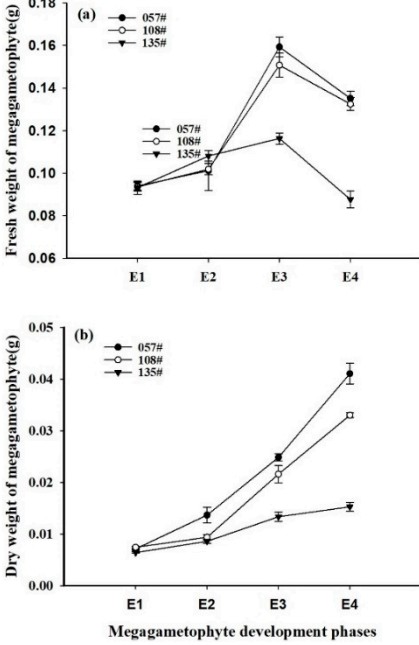

**Figure 2.** (**a**) Fresh weight of megagametophyte (**b**) dry weight of megagametophyte in different developmental stages of *Pinus koraiensis*. Each data point was repeated for 10 samples. The error bars represent the standard error.

3.1.2. EC Induction

The explants at the E1 stage were placed onto the induction medium, and cell proliferation was initiated at the micropylar end of the megagametophyte within 30 days of culturing (Figure 3a). The cytological observation indicated that there were vacuolated cells (Figure 3b), embryogenic cell groups (Figure 3c), symmetrical embryogenic cells (Figure 3d) asymmetrical embryogenic cells

(Figure 3e) and early SEs (Figure 3f) in the EC. Some of the embryogenic cells divided symmetrically (Figure 3d), which made the EC proliferate continuously. At the same time, some of the embryogenic cells divided asymmetrically (Figure 3e) and gradually developed into early SEs (Figure 3f).

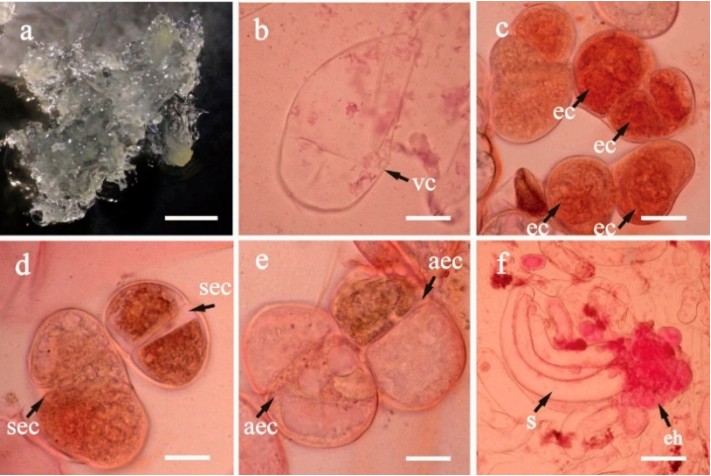

**Figure 3.** Cytological observation of the EC of Pinus koraiensis after safranine staining: (**a**) EC, bar = 0.5 cm; (**b**) Vacuolated cells (vc), bar = 20 μm; (**c**) Embryogenic cell group, bar = 20 μm; (**d**) Symmetrical embryogenic cells (sec), bar = 20 μm; (**e**) Asymmetrical embryogenic cells (aec), bar = 20 μm; (**f**) Early SEs, bar = 60 μm; The letters 'eh' denote the embryo head (eh), and the letter 's' denotes the suspensor (s).

The induction percentage of EC differed significantly among different families ($p < 0.05$). In 2015, the highest induction percentage was in family 135#, followed by family 108# and then 057# (Figure 4). The average induction percentage of EC varied greatly among the different cone collection times. The EC induction percentage of E1 was the highest, followed by E2, E3 and then E4.

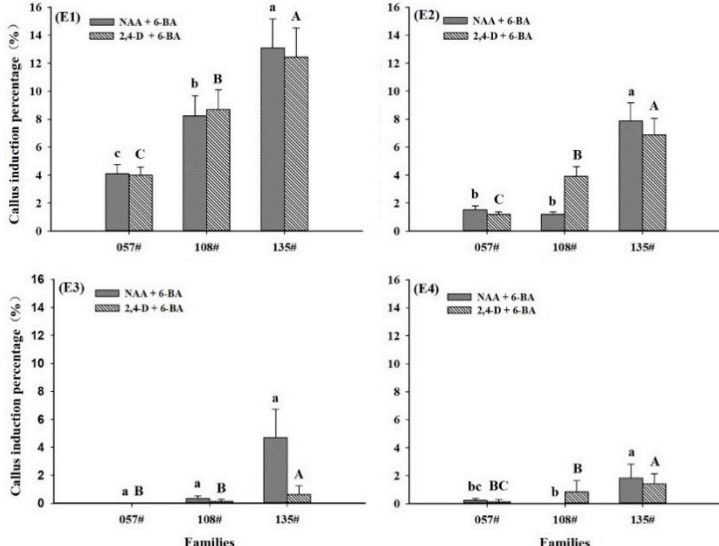

**Figure 4.** Comparison of EC induction percentage of megagametophytes in three Pinus Koraiensis families at different development stages. Each data point was repeated for 16 replicates. The error bars represent the standard error. The different small letters in the figure show a significant difference at $p < 0.05$ on the medium of NAA + 6-BA combination, and different capital letters show a significant difference at $p < 0.05$ on the medium of 2,4-D + 6-BA combination.

Megagametophytes at the E1 stage from the 135# family were used as the explant. Among the media with combinations of NAA and 6-BA, the optimal medium for EC induction was DCR + 35 g L$^{-1}$ sucrose + 10.74 μmol·L$^{-1}$ NAA + 6.66 μmol·L$^{-1}$ 6-BA + 0.8 mg L$^{-1}$ CH, with the highest induction percentage of 33.33% (Table 1). Among the media with combinations of 2,4-D + 6-BA, the optimal medium was DCR + 35 g L$^{-1}$ sucrose + 27.14 μmol·L$^{-1}$ 2,4-D + 6.66 μmol·L$^{-1}$ 6-BA + 0.8 g L$^{-1}$ CH, and the maximum induction percentage was 31.90% (Table 2). Across all media (combinations of NAA + 6-BA and 2,4-D + 6-BA), the factors could be ranked, from strongest influence on the induction percentage to weakest, as follows: NAA/2,4-D > sucrose > 6-BA > CH.

**Table 2.** Influencing factors of source, acid-hydrolyzed casein (CH), 2,4-dichlorophenoxyacetic acid (2,4-D) and N$^6$-benzyladenine (6-BA) on embryogenic lines (EC) induction of *Pinus koraiensis* based on orthogonal experiments design.

| Treatments | Sucrose Concentration (g L$^{-1}$) | 2,4-D Concentration (μmoL·L$^{-1}$) | 6-BA Concentration (μmoL·L$^{-1}$) | CH (g L$^{-1}$) | Total Explants | Induction Percentage (%) |
|---|---|---|---|---|---|---|
| 1 | 25 | 18.1 | 2.22 | 0.3 | 96 | 11.65 ± 0.90 |
| 2 | 25 | 27.14 | 4.44 | 0.5 | 127 | 10.32 ± 1.33 |
| 3 | 25 | 36.2 | 6.66 | 0.8 | 69 | 11.66 ± 1.85 |
| 4 | 25 | 45.24 | 8.88 | 1.0 | 81 | 0.00 ± 0.00 |
| 5 | 30 | 18.1 | 4.44 | 0.8 | 72 | 12.57 ± 1.49 |
| 6 | 30 | 27.14 | 2.22 | 1.0 | 87 | 25.32 ± 1.51 |
| 7 | 30 | 36.2 | 8.88 | 0.3 | 103 | 11.63 ± 1.23 |
| 8 | 30 | 45.24 | 6.66 | 0.5 | 113 | 13.37 ± 1.00 |
| 9 | 35 | 18.1 | 6.66 | 1.0 | 102 | 23.54 ± 1.37 |
| 10 | 35 | 27.14 | 8.88 | 0.8 | 116 | 31.90 ± 1.23 |
| 11 | 35 | 36.2 | 2.22 | 0.5 | 112 | 12.42 ± 1.06 |
| 12 | 35 | 45.24 | 4.44 | 0.3 | 117 | 4.35 ± 1.28 |
| 13 | 40 | 18.1 | 8.99 | 0.5 | 130 | 11.60 ± 1.22 |
| 14 | 40 | 27.14 | 6.66 | 0.3 | 104 | 8.64 ± 1.21 |
| 15 | 40 | 36.2 | 8.88 | 1.0 | 119 | 9.29 ± 1.28 |
| 16 | 40 | 45.24 | 2.22 | 0.8 | 153 | 0.65 ± 0.65 |

Note: The values in the column Induction Percentage in the table represent the mean ± standard error.

In 2017, EC was produced from explants from 6 of the 24 families; the highest induction percentage was 12.00% and the lowest was 2.00% The induction percentage for other families was zero, and the induction percentage differed significantly among family sources (Figure 5).

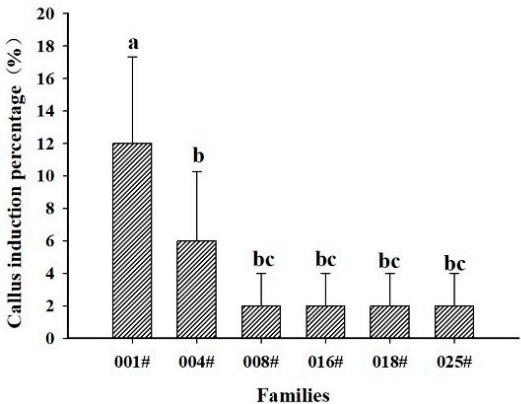

**Figure 5.** Explant EC induction results from different sources families of *Pinus koraiensis*. Each data point was repeated for 10 replicates. The error bars represent the standard error. The different small letters in the figure show a significant difference at *p* < 0.05.

### 3.2. Proliferation and Maturation of EC

#### 3.2.1. Proliferation Experiment 1: Effect of PGRs on the Proliferation and Maturation of Korean Pine EC

The proliferation efficiency of three cell lines was significant ($p < 0.05$) under different PGRs concentrations (Table 3), and it was increased with the increase of PGRs concentration. The combination of 9.04 $\mu mol \cdot L^{-1}$ 2,4-D and 4.4 $\mu mol \cdot L^{-1}$ 6-BA had the strongest proliferation ability (cell line 001#-100 was 617.00%), followed by 4.52 $\mu mol \cdot L^{-1}$ 2,4-D + 2.2 $\mu mol \cdot L^{-1}$ 6-BA, and finally 2.26 $\mu mol \cdot L^{-1}$ 2,4-D + 0.44 $\mu mol \cdot L^{-1}$ 6-BA.

**Table 3.** Effects of PGRs concentration on EC proliferation of *Pinus koraiensis*.

| Treatments | PGRs Concentration ($\mu mol \cdot L^{-1}$) | | Proliferation Efficiency (%) | | |
|---|---|---|---|---|---|
| | 2,4-D | 6-BA | 001#-100 | 001#-1 | 001#-34 |
| 1 | 9.04 | 4.4 | 617.00 ± 31.33 a | 524.00 ± 29.77 a | 605.00 ± 37.52 a |
| 2 | 4.52 | 2.2 | 483.00 ± 38.78 b | 414.00 ± 25.47 b | 496.00 ± 31.20 ab |
| 3 | 2.26 | 0.44 | 414.00 ± 29.97 b | 388.00 ± 23.64 b | 461.00 ± 39.03 b |

Note: Each cell line with five replicates. The values in column Proliferation Efficiency in the table represent the mean ± standard error. Different letters in the same column indicate a significant difference at $p < 0.05$.

The microstructure of the paraffin section is shown in Figure 6. The late stage of SEs gradually developed after 10 days of mature culturing (stage I, SEs with elongated suspensors, Figure 6a,d), stage II developed at about 30 days (precotyledon embryo, Figure 6b,e), and stage III developed at about 45 days (cotyledonary embryo, Figure 6c,f). There were significant differences in the development and maturation abilities of proliferative SEs under three PGRs concentrations ($p < 0.05$) (Table 4), and the ability in different cell lines was also different. Under the combination of 9.04 $\mu mol \cdot L^{-1}$ 2,4-D and 4.4 $\mu mol \cdot L^{-1}$ 6-BA, the proliferation efficiency of the embryogenic cell line was the best, but the development and maturity ability of SEs was the worst (cell line 001#-034 was 17.86 $\cdot g^{-1}$ FW). However, under the combination of 4.25 $\mu mol \cdot L^{-1}$ 2,4-D and 2.2 $\mu mol \cdot L^{-1}$ 6-BA, the proliferation efficiency of the embryogenic cell line was less than with the combination of 9.04 $\mu mol \cdot L^{-1}$ 2,4-D and 4.4 $\mu mol \cdot L^{-1}$ 6-BA, but the development and maturation ability of SEs was the best (cell line 001#-100 was 135.71 $\cdot g^{-1}$ FW).

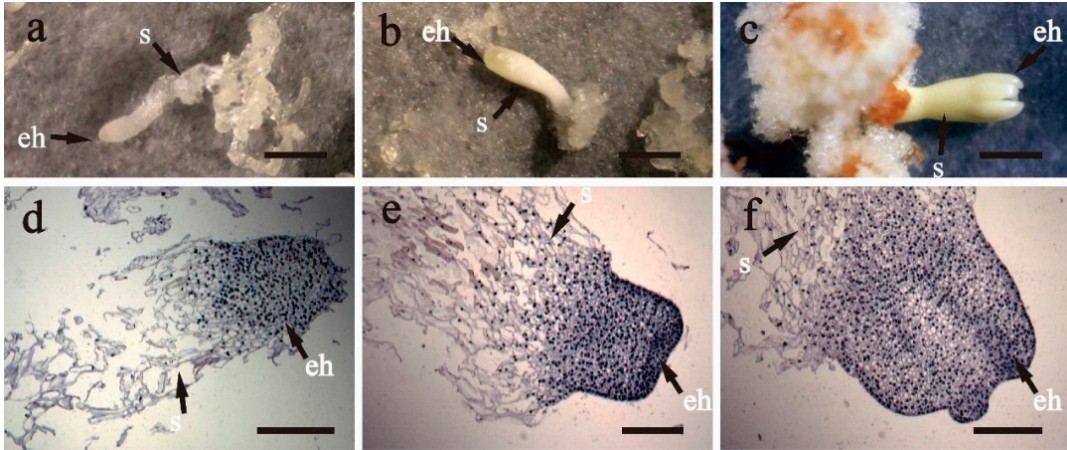

**Figure 6.** Morphological and paraffin section observation of the mature process of *Pinus koraiensis* SEs. (**a**) Late SEs stage I (stage I), bar = 0.1 cm; (**b**) Late SEs stage II (stage II), bar = 0.2 cm; (**c**) Late SEs stage III (stage III), bar = 0.2 cm; (**d**) Paraffin section of late SEs stage I, bar = 200 µm; (**e**) Paraffin section of late SEs stage II bar = 100 µm; (**f**) Paraffin section of late SEs stage III bar = 100 µm; The letters 'eh' denote the embryo head (eh), and the letter 's' denotes the suspensor 's'.

**Table 4.** Effects of PGRs concentration on the *Pinus koraiensis* SEs' development and maturation.

| Treatments | PGRs Concentration ($\mu mol \cdot L^{-1}$) | | Number of Cotyledon SEs of 3 Cell Lines (No./g$^{-1}$FW) | | |
|---|---|---|---|---|---|
| | 2,4-D | 6-BA | 001#-100 | 001#-001 | 001#-034 |
| 1 | 9.04 | 4.4 | 95.83 ± 7.68 b | 97.50 ± 11.46 a | 17.86 ± 5.36 b |
| 2 | 4.52 | 2.2 | 135.71 ± 13.48 a | 118.75 ± 13.21 a | 37.50 ± 6.46 a |
| 3 | 2.26 | 0.44 | 100.00 ± 13.31 ab | 91.67 ± 7.68 a | 27.08 ± 3.84 ab |

Note: Each cell line with six replicates. The values in column Number of Cotyledon SEs of 3 Cell Lines in the table represent the mean ± standard error. Data represent the number of SEs formed per gram of fresh weight, and different letters in the same column indicate a significant difference at $p < 0.05$.

### 3.2.2. Proliferation Experiment 2: The Effect of L-Glutamine on the Proliferation and Maturation of Korean Pine EC

Different L-glutamine concentrations had no significant effect on the proliferation efficiency of EC (Table 5) and cell structure (Figure 7a,b), but had a significant effect on the development and maturation ability of SEs ($p < 0.05$) (Table 6). When EC proliferated at 0.5 g L$^{-1}$ L-glutamine, the number of late stage embryos in stage I was less than 0.5 g L$^{-1}$ L-glutamine (Figure 7c,d), as was the number of mature embryos of stage III (cell line 001#-100 was the highest, which was 118.75·g$^{-1}$ FW). In addition, when the concentration of L-glutamine was increased to 1 or 1.5 g L$^{-1}$, the development and maturation of the SEs of the three cell lines were improved, but the difference was not significant. When EC proliferated under the condition of 1.5 g L$^{-1}$ L-glutamine, the number of late stage embryos of stage I was more than 0.5 g L$^{-1}$ L-glutamine (Figure 7c,d, 001#-001 cell line had the best ability of embryo development and maturation). Furthermore, when under the condition of 1 g L$^{-1}$ L-glutamine, 001#-100 and 001#-034 cell lines had the best capacities for SEs maturation (cell line 001#-100 was 165.63·g$^{-1}$ FW, 001#-034 was 46.88·g$^{-1}$ FW).

**Table 5.** Effects of L-glutamine concentration on the EC proliferation of *Pinus koraiensis*.

| Treatments | L-glutamine Concentration (g L$^{-1}$) | Proliferation Efficiency (%) | | |
|---|---|---|---|---|
| | | 001#-100 | 001#-100 | 001#-034 |
| 1 | 0.5 | 467.50 ± 44.84 a | 410.00 ± 30.62 a | 492.50 ± 27.73 a |
| 2 | 1 | 481.25 ± 48.41 a | 396.25 ± 20.14 a | 487.50 ± 30.10 a |
| 3 | 1.5 | 455.00 ± 52.08 a | 408.75 ± 43.22 a | 455.00 ± 56.20 a |

Note: Each cell line with four replicates. The values in column Proliferation Efficiency in the table represent the mean ± standard error. Different letters in the same column indicate a significant difference at $p < 0.05$.

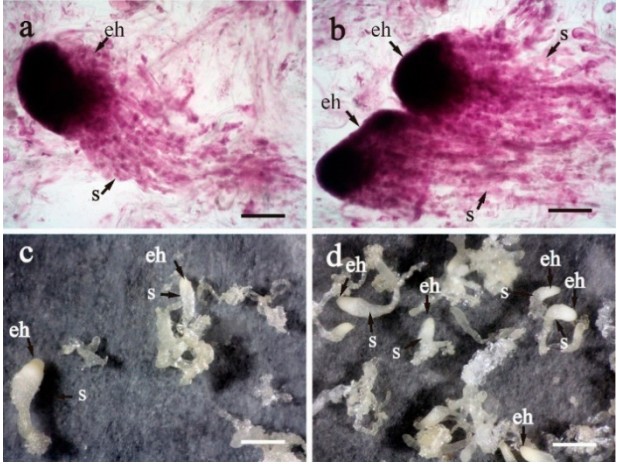

**Figure 7.** Cell viability of EC of *Pinus koraiensis* under different L-glutamine conditions. (**a**) EC at 0.5 g L$^{-1}$ L-glutamine, bar = 100 μm; (**b**) EC at 1.5 g L$^{-1}$ L-glutamine, bar = 100 μm; (**c**) Late SEs with poor viability, bar = 0.2 cm; (**d**) Late SEs with good viability, bar = 0.2 cm. The letters 'eh' denote the embryo head 'eh', and the letter 's' denotes the suspensor 's'.

**Table 6.** Effects of L-glutamine concentration on the *Pinus koraiensis* SEs development and maturation.

| Treatments | L-glutamine Concentration (g L$^{-1}$) | Number of SEs of 3 Cell Lines (No./g$^{-1}$FW) | | |
|:---:|:---:|:---:|:---:|:---:|
| | | 001#-100 | 001#-001 | 001#-034 |
| 1 | 0.5 | 118.75 ± 10.83 b | 103.13 ± 7.86 b | 28.13 ± 9.38 a |
| 2 | 1 | 165.63 ± 5.98 a | 146.88 ± 5.98 a | 46.88 ± 7.86 a |
| 3 | 1.5 | 162.50 ± 5.10 a | 162.50 ± 5.10 a | 43.75 ± 6.25 a |

Note: Each cell line with four replicates. The values in column Number of Cotyledon SEs of 3 Cell Lines in the table represent the mean ± standard error. Data represent the number of SEs formed per gram of fresh weight, and different letters in the same column indicate a significant difference at $p < 0.05$.

### 3.3. SEs Germination and Plant Regeneration

The development and maturation ability of 001#-100 SEs was the best (Table 6). However, some of the SEs germinated normally (Figure 8a, letter n), and some of them abnormally (Figure 8a, letter a). As a whole, the germination ability of the 001#-100 SEs was the worst (36.00%) (Table 7). Among the three cell lines tested, 001#-034 had the strongest germinating ability (66.00%), followed by 001#-001 (58.00%) and finally 001#-100. After 16 weeks of SEs germination and culture, they were transplanted into the substrate (nutrient soil: vermiculite: Perlite = 3:1:1) (Figure 8b).

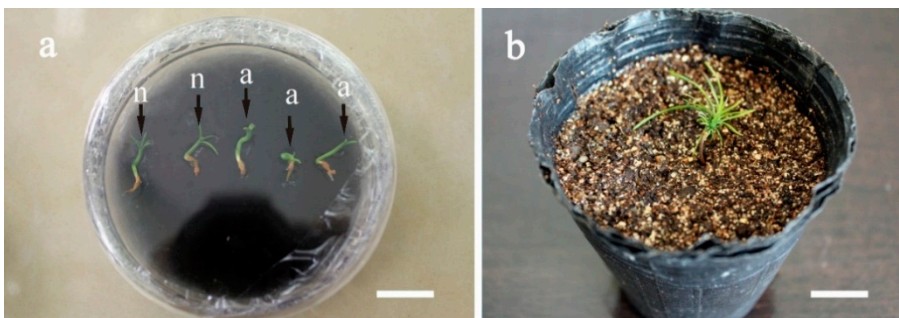

**Figure 8.** SEs germination and plant regeneration of *Pinus koraiensis*. (**a**) Germination of SEs, bar = 2 cm; (**b**) Regenerated plant, bar = 2 cm; The letter 'n' indicates normal germination of SEs, and the letter 'a' indicates abnormal germination of SEs.

**Table 7.** Germination of mature SEs of *Pinus koraiensis*.

| Cell Lines | Number of SEs (number) | Normal SEs Percentage (%) |
|:---:|:---:|:---:|
| 001#-100 | 50 | 36.00 ± 5.10 |
| 001#-001 | 50 | 58.00 ± 5.83 |
| 001#-034 | 50 | 66.00 ± 8.72 |

Note: The values in column Normal SEs Percentage in the table represent the mean ± standard error.

## 4. Discussion

### 4.1. Initiation of EC

The induction percentage of coniferous EC is usually very low, and depends heavily on the physiological stage of the explant, the location of the explant, the genotype of the families, the nutrient composition of the medium, the types and concentrations of growth regulators, and the culture conditions [23]. Our previous research shows that, as the age and physiological status of the mother tree increases, the callus induction percentage decreases significantly [19]. In most previous studies, EC has been induced from young tissues of coniferous species [26]. Many studies have reported that the EC induction percentages of Pinus and Picea are higher at the cleavage polyembryonic and prophase of cotyledon embryo stages than at other stages. In the present study, the best stage for the EC induction of *P. koraiensis* was the E1 stage (proembryo stage). The EC induction percentage

was lower at the E3 and E4 stages than at the E1 stage, different from most other conifer species [23]. Our previous research shows that the highest induction percentage of EC was 1.67% [19]. In this study, the highest induction percentage was 33.33%, the lowest was 0.14%, and the induction percentage of EC was significantly increased. We found that there were significant differences in EC induction percentage among different families, like in our previous research [19]. The induction percentages of 24 families in 2017 were lower than those in 2015 (the highest induction percentage is 12%), and this may be related to the genotype of the families, or to the unsuitable development of immature seeds. Further verification is required in the later test process.

Phytohormones are the key substances controlling the process of somatic embryogenesis in culture in vitro. The induction and proliferation of the embryogenic cultures of many conifers proceed on standard nutrient medium supplemented with exogenous PGRs [27]. Auxin and cytokinin are used at the induction stage of Pinus [6]. In previous studies, the most commonly used auxin was 2,4-D, and the most commonly used cytokinin was 6-BA [28]. Auxin is considered to be the most critical factor in the induction stage [26]. In this study, combinations of NAA and 6-BA, or 2,4-D and 6-BA, at different concentrations were used to induce EC from explants of *P. koraiensis*. Both sets of combinations induced EC of *P. koraiensis*. Overall, the induction percentage was slightly higher with combinations of NAA and 6-BA than with combinations of 2,4-D and 6-BA, which was different from most other pine species [29–31].

### 4.2. Multiplication and Maturation of EC

EC is the foundation of regenerating plants on a large scale, and serves as an important material in genetic transformation. It is also an ideal system for studying the entirety of single-cell differentiation and the expression of totipotency [19]. The quality of EC not only affects the proliferation efficiency, but also affects the quantity and quality of SEs [32,33]. The quality of SEs is also a key factor in evaluating the success of somatic embryogenesis. In our study, the PGRs concentration significantly affected the EC proliferation and maturation, while L-glutamine concentration had no significant effect on the proliferation efficiency, but had a significant effect on the SEs' maturation. Therefore, during somatic embryogenesis, not only the proliferation efficiency but also the development and maturity ability of SEs should be considered.

### 4.3. Germination of Mature SEs

Among these phases, germination/conversion is regarded as the most important step in obtaining plantlets; this determines the success of this technique. Morphologically, mature conifer somatic embryos cannot germinate or convert into viable plantlets unless the embryos undergo partial desiccation treatment. This treatment has been used effectively to improve the germination/conversion of somatic embryos [34,35]. When SE was partially desiccated, the highest germination percentage was 66%. In the process of SEs germination, some SEs germinate abnormally [32,36,37], which limits the process of large-scale breeding. In this study, we found that the germinating ability of SEs of different genotypes was different. In the next step, we need to further optimize the key technology of EC proliferation of Korean pine, prolong the retention time of EC, enhance the ability of embryo maturation and germination, and lay the foundation for large-scale propagation.

## 5. Conclusions

In summary, this work demonstrates that efficient SE induction and establishment in continental and Mediterranean provenances of Korean pine depends not only on the mother tree but also on the developmental stage of the megagametophyte. Culture conditions during the EC proliferation stage significantly affect the maturation of SEs. Therefore, during somatic embryogenesis, not only the proliferation efficiency but also the development and maturity ability of SEs should be considered. We established systems for EC induction, somatic embryogenesis, plant regeneration

and cryopreservation. These systems can be used to propagate the excellent germplasm resources of *P. koraiensis*, and for the establishment of multi-varietal forestry.

**Author Contributions:** L.Y. and H.S. conceived and designed the study. F.G. and C.P. collected plant materials and prepared SE samples for analysis. F.G. analyzed the results for experiments. L.Y., F.G. and H.W. wrote the paper. I.N.T. and A.M.N. revised the manuscript. All authors have read and agreed to the published version of the manuscript.

**Funding:** The work was supported by the National Key R&D Program of China (2017YFD0600600), and the Innovation Project of State Key Laboratory of Tree Genetics and Breeding (Northeast Forestry University, 2016C01).

**Acknowledgments:** We thank two anonymous reviewers and the editor for comments that improved an earlier draft of this article.

**Conflicts of Interest:** The authors declare no conflict of interest.

## Abbreviations

| | |
|---|---|
| 2,4-D | 2,4-dichlorophenoxyacetic Acid |
| 6-BA | N6-benzyladenine |
| ABA | Abscisic acid |
| CH | Acid-hydrolyzed casein |
| DCR | Basal culture medium (Gupta and Durzan 1985) |
| DMSO | Dimethyl sulfoxide minimum |
| mLV | Litvay medium (Litvay et al. 1985, modified by Hargreaves et al., 2009) |
| NAA | 1-Naphthalacetic Acid |
| SEs | Somatic embryos |
| EC | Embryogenic lines |
| PGRs | Plant growth regulator |

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
