# Peer review of "Key Techniques for Somatic Embryogenesis and Plant Regeneration of Pinus koraiensis"

_forests, doi:10.3390/f11090912_

Round 1

Reviewer 1 Report

Key techniques for somatic embryogenesis and plant regeneration of Pinus koraiensis

Fang Gao, Chunxue Peng, Hao Wang, Ling Yang and Hailong Shen

Korean pine is an economically-important tree species in north-eastern Asia that has recently suffered a sharp decline in China. The authors therefore investigated ways to propagate this plant via somatic embryogenesis. They tested effects of collection time and various combinations of growth regulators and other medium components on the induction of embryogenic lines from seeds collected from 27 families and their subsequent cryopreservation. They found that mLV medium supplemented with 30 g L-1 sucrose, 4.52 μmol·L-1 2,4-D, 2.2 μmol· L-1 6-BA, 0.5 g L-1 CH, 1.5 g L-1 L-glutamine, and 4 g L-1 Gelrite gave the best results. They obtained 3 cell lines that gave up to 66% germination, thus showing the feasibility of this approach.

Overall I think that this study has been properly performed with sufficient replication, and provides useful data worth sharing with the plant biology community. Although some of it is highly specific to Korean pine, it will form a useful starting point for studies with other species and the approach used will also be a useful template for researchers attempting similar studies in other plant species.

I am troubled by the use of synthetic growth regulators, especially 2,4-D, since it has been shown to have different effects than natural auxins such as IAA or IBA. Similarly, I am concerned by the use of 6-BA, although not as much. I concede that the authors were successful in regenerating plants using these synthetic growth regulators, but I wonder what would happen if the authors were to use naturally occurring auxins such as IAA or IBA and naturally occurring cytokinins, such as zeatin or kinetin.

The English is usually understandable, but many if not most sentences contain errors that require attention. However, here are some errors that should be fixed

  • “Casein acid hydrolyzed” should be “acid-hydrolyzed casein”
  • The caption to figure 2 should indicate the number of samples for each data point and whether the error bars represent the standard deviation or the standard error.
  • The caption to figure 4 should indicate the number of samples for each column and whether the error bars represent the standard deviation or the standard error.
  • The title of Table 1 is hard to understand and should be rewritten. What do they mean by “Influencing factors of L (4)4 …”
  • The title of Table 2 is also hard to understand and should be rewritten. What do they mean by “Influencing factors e of L (4)4 …”
  • The caption to figure 4 should indicate the number of samples for each column and whether the error bars represent the standard deviation or the standard error. They should also explain the letters above each error bar.
  • Tables 3, 4, 5 and 6 should indicate the number of samples and whether the error is standard deviation or standard error.

Author Response

Response to Reviewer 1 Comments

Dear Reviewer,

Our sincere thanks to you for the time and effort that you have put into reviewing our manuscript! We found all the comments very constructive and helpful, and have revised our manuscript according to all comments. Please find, below, our point-by-point response to the comments raised.

Thank you for considering our revised manuscript!

Point 1: I am troubled by the use of synthetic growth regulators, especially 2,4-D, since it has been shown to have different effects than natural auxins such as IAA or IBA. Similarly, I am concerned by the use of 6-BA, although not as much. I concede that the authors were successful in regenerating plants using these synthetic growth regulators, but I wonder what would happen if the authors were to use naturally occurring auxins such as IAA or IBA and naturally occurring cytokinins, such as zeatin or kinetin.

Response 1: Our previous research proved:The callus did not remain embryogenic over the long-term on media containing IBA + 6-BA, IBA + ABA, 6-BA + KT, or 2,4-D + ABA. Moreover, callus readily browned in media with IBA + ABA and 2,4-D + ABA. With 2,4-D + 6-BA, embryogenicity was maintained, and cells proliferated for a long time (Gao et al, 2020. J. For. Res).

Point 2:The English is usually understandable, but many if not most sentences contain errors that require attention. However, here are some errors that should be fixed. “Casein acid hydrolyzed” should be “acid-hydrolyzed casein”

Response 2: Line 27 ‘Casein acid hydrolyzed’ has been replaced by ‘acid-hydrolyzed casein’

Point 3:The caption to figure 2 should indicate the number of samples for each data point and whether the error bars represent the standard deviation or the standard error.

Response 3: Line 224 Each data point was repeated 10 samples. The error bars represent the standard error.

Point 4:The caption to figure 4 should indicate the number of samples for each column and whether the error bars represent the standard deviation or the standard error.

Response 4: Line 245-249 Each data point was repeated 16 replications. The error bars represent the standard error.  

Point 5:The title of Table 1 is hard to understand and should be rewritten. What do they mean by “Influencing factors of L (4)4 …”.

Response 5: Line 258-260 Table 1. Influencing factors of Source, Acid-hydrolyzed casein (CH), 1-Naphthalacetic Acid (NAA) and N6-benzyladenine (6-BA) on Embryogenic lines (EC) induction of Pinus Koraiensis based on orthogonal experiments design

Point 6: The title of Table 2 is also hard to understand and should be rewritten. What do they mean by “Influencing factors e of L (4)4 …”

Response 6: Line 262-264 Table 2. Influencing factors of Source, Acid-hydrolyzed casein (CH), 2,4-dichlorophenoxyacetic acid (2,4-D) and N6-benzyladenine (6-BA) on Embryogenic lines (EC) induction of Pinus Koraiensis based on orthogonal experiments design

Point 7: The caption to figure 4 should indicate the number of samples for each column and whether the error bars represent the standard deviation or the standard error. They should also explain the letters above each error bar.

Response 7: Line 245-249 Note: Each data point was repeated 16 replications. The error bars represent the standard error. The different small letters in the figure show significant difference at 0.05 on the medium of NAA + 6-BA combination, and different capital letters show significant difference at 0.05 on the medium of 2,4-D + 6-BA combination.

Point 8: Tables 3, 4, 5 and 6 should indicate the number of samples and whether the error is standard deviation or standard error.

Response 8: Line 282-284 Tables 3:Note: Each cell line with 5 replications. The values in column proliferation efficiency in the table represent the mean ± standard error.

Tables 4:Line 303-305 Note: Each cell line with 6 replications. The values in column number of cotyledon SEs of 3 cell lines in the table represent the mean ± standard error.

Tables 5:Line321-322 Note: Each cell line with 4 replications. The values in column proliferation efficiency in the table represent the mean ± standard error. 

Tables 6:Note: Each cell line with 4 replications. The values in column number of cotyledon SEs of 3 cell lines in the table represent the mean ± standard error.

Reviewer 2 Report

Dear Authors,

The work You presented fits with the journal’s scope and results presented here are novel and implemented previous protocols on somatic embryogenesis in K. pine. However, some points need to be clarified. In particular, my suggestion is to short cut the paper, deleting the cryopreservation: there are no evidence of results. No pictures, no data, except of “all of them survived” (line 323)

Then, some paragraphs should be implemented.

Please consider some suggestion prior to publish Your work:

Introduction:

  1. Literature must be implemented with your previous publication on Korean pine (Gao et al, 2020. J. For. Res) and other works on this species, to let people better understand what improvement you have done.
  2. Bibliography: some sentences need only one reference (for example line 55, line 63….) and you can delete some references not so specific by citing a valuable review in conifers proposed by Klimaszewska, Krystyna (et al.) “Advances in Conifer Somatic Embryogenesis Since Year 2000” .
  3. Line 70: this statement is not clear/incomplete: I suggest to change in “most Pinus species are recalcitrant to micropropagate and regenerate by SE; Korean pine is not an exception” (please provide references)
  4. Line 70-75: It would help to clarify what Bozhkov found, and regarding the loss of ability to proliferate over-time please provide the right references.
  5. Cryopreservation: this sound as a new paragraph. However, in my opinion one sentence could be sufficient, due to the fact that results provided in this paper are not robust
  6. Line 79: I supposed that “seeds from” 27 families were collected. Then correct the punctuation.

Mat and meths:

Paragraph 2.1

  1. my suggestion is calling this section “EC induction” and describe first the explant source, then the collection time (2015, 2017; E1 or E2 phases etc…), how many families were treated and how many lines were chosen for the further steps.
  2. please clarify where the orchard stands
  3. please define the stage of the somatic embryos (i.e. zygotic-derived embryo, stage E1) here or in the introduction, and not only in the “Results” paragraph
  4. Please correct “replications” with replicates (here and in other paragraphs)
  5. I cannot understand the sentence “The best scheme selected in 2015 was used for EC…”: please could You clarify this paragraph? Based on the M&M description, one cannot know if there are more collecting seasons (as the results suggest: 2015 and 2017).

Paragraph 2.2

  1. mLV medium : please provide only one citation
  2. I suggest to split in two different paragraph experiments on proliferation and maturation: for the last step, which is the same for both the proliferation experiments, You can add a single paragraph.
  3. Cryopreservation: there are no evidence of results in cryopreservation experiment. Hence my suggestion is to delete this paragraph and eventually put some comments on “discussion”. Otherwise, in “results”, you should write “data not shown”.

Paragraph 2.3

As I stated before, this paragraph could be deleted because there are no robust results about cryopreservation. Otherwise, you can just cite Reeve’s method, and clarify how “recovery vigour test” (line 161.) have done.

Paragraph 2.4

style must be revised: it looks like a copy and paste of a bench protocol (from line 169 on).

English must be revised: the SEs needed to be partial desiccated before germination; mLV medium supplemented WITH 2 g ….; embryos were PLACED and not inoculated onto germination medium.

 I also suggest to replace the world “inoculated” with the correct word “placed” throughout all the text.

Results:

I cannot understand if fig. 1 refers to 2015 or 2017

at line 258 You stated “24 families”, but in the introduction You stated 27. Which data is correct?

Line282: “the proliferation efficiency of embryogenic cell line was general”: please clarify this sentence

Paragraph 3.2.2. Each time You state “was less” or “was more” please make a comparison: less than…; more than…

Paragraph 3.3. provide evidences or delete this paragraph.

Pictures/tables: pictures and tables are nice and clearly describes the results, however there are some typos in the captions, please check them.

Discussion:

4.1 Please discuss the improvement/difference with your previous work in Korean p0ine (Gao et al 2020) and provide reference at row 345.

4.2 You can incorporate one-two sentences in the previous paragraph.

4.3 line 389: change the sentence with “When SE were partially desiccated, highest germination percentage was 66%”

As a general suggestion, punctuation must be revised (many sentences started with “and”, commas and semi-colon needs an extensive English review) and English extensively revised. Further, paragraph in math and methods and results should follow the same order.

Author Response

Response to Reviewer 2 Comments

Dear Reviewer,

Our sincere thanks to you for the time and effort that you have put into reviewing our manuscript! We found all the comments very constructive and helpful, and have revised our manuscript according to all comments. Please find, below, our point-by-point response to the comments raised.

Thank you for considering our revised manuscript!

Point 1: The work You presented fits with the journal’s scope and results presented here are novel and implemented previous protocols on somatic embryogenesis in K. pine. However, some points need to be clarified. In particular, my suggestion is to short cut the paper, deleting the cryopreservation: there are no evidence of results. No pictures, no data, except of “all of them survived” (line 323)

Response 1: We have deleted the part about cryopreservation.

Point 2: Literature must be implemented with your previous publication on Korean pine (Gao et al, 2020. J. For. Res) and other works on this species, to let people better understand what improvement you have done. 1.     

Response 2: Line 81 We have improved the problems in the literature and improved the solution.

Point 3: Bibliography: some sentences need only one reference (for example line 55, line 63….) and you can delete some references not so specific by citing a valuable review in conifers proposed by Klimaszewska, Krystyna (et al.) “Advances in Conifer Somatic Embryogenesis Since Year 2000” . 2.

Response 3: Line 83 We have deleted less specific articles and cited articles published by Klimaszewska et al, 2016.

Point 4: Line 70: this statement is not clear/incomplete: I suggest to change in “most Pinus species are recalcitrant to micropropagate and regenerate by SE; Korean pine is not an exception” (please provide references)

Response 4: Line 78 We revised the sentences in the article based on the reviewers' comments and cited references. Modify as follows:

‘Most Pinus species are recalcitrant to micropropagation and regeneration by SE[20]. and Korean pine is not an exception’

Point 5: Line 70-75: It would help to clarify what Bozhkov found, and regarding the loss of ability to proliferate over-time please provide the right references.     

Response 5: Line 79 We have inserted references

Point 6:  Cryopreservation: this sound as a new paragraph. However, in my opinion one sentence could be sufficient, due to the fact that results provided in this paper are not robust

Response 6: We have deleted the part about cryopreservation in the article.

Point 7: Line 79: I supposed that “seeds from” 27 families were collected. Then correct the punctuation.

Response 7: Line 86 ‘27 families were collected using immature megagametophytes as the explant’ has been replaced by ‘seeds from 27 families were collected. Using immature megagametophytes as the explant’

Mat and meths:

Paragraph 2.1

Point 8: my suggestion is calling this section “EC induction” and describe first the explant source, then the collection time (2015, 2017; E1 or E2 phases etc…), how many families were treated and how many lines were chosen for the further steps.

Response 8: Line 94-104 We have modified this part. Modify as follows:

From June to July 2015, open pollinated cones of three families (numbered 057#, 108#, and 135#) were collected from the Qingshan Forest Seed Orchard, Weihe Forestry Bureau, Heilongjiang Province, China. The age of each family was 28 years, and the date of seed collection was June 23rd, June 30th, July 6th, and July 13th (representing seeds in the E1, E2, E3, and E4 phases, respectively). The cones were cut from the branches and stored at 4°C. The physiological state of explants from seeds collected at the four times was determined by observation under a microscope (Olympus BX51, Olympus Corp., Tokyo, Japan) and photographed using a Moticam 3000C camera (Seneco, Milan, Italy). At the same time, ten seeds were randomly selected, and the fresh weight of megagametophyte was measured by peeling off the seed coat, then put them in the oven at 108°C to constant weight, and measured the dry weight.

On July 1st, 2017, open-pollinated cones in the E1 phase were collected from 24 families, and their megagametophytes were used as explants.

Point 9: please clarify where the orchard stands

Response 9: Line 94-104 We have modified this part.

Point 10: please define the stage of the somatic embryos (i.e. zygotic-derived embryo, stage E1) here or in the introduction, and not only in the “Results” paragraph

Response 10: Line 97 We have modified this part. Modify as follows:

(representing seeds in the E1, E2, E3, and E4 phases, respectively)

Point 11: Please correct “replications” with replicates (here and in other paragraphs)

Response 11: Line 123 ‘replications’ has been replaced by ‘replicates’

Point 12: I cannot understand the sentence “The best scheme selected in 2015 was used for EC…”: please could You clarify this paragraph? Based on the M&M description, one cannot know if there are more collecting seasons (as the results suggest: 2015 and 2017).

Response 12: Line 94-128 We have indicated in the article the collection time and culture conditions of the cones of the 24 families in 2017 based on the comments of the reviewers, see 2.1.1 and 2.1.3 for details.

Point 13: mLV medium : please provide only one citation

Response 13: Line 143 We have modified the cited references and only kept one reference.

Point 14: I suggest to split in two different paragraph experiments on proliferation and maturation: for the last step, which is the same for both the proliferation experiments, You can add a single paragraph. Response 14: We have adjusted the paragraphs in the materials and methods.

Point 15: Cryopreservation: there are no evidence of results in cryopreservation experiment. Hence my suggestion is to delete this paragraph and eventually put some comments on “discussion”. Otherwise, in “results”, you should write “data not shown”.

Response 15: We have deleted content related to cryopreservation.

Point 16: As I stated before, this paragraph could be deleted because there are no robust results about cryopreservation. Otherwise, you can just cite Reeve’s method, and clarify how “recovery vigour test” (line 161.) have done.

Response 16: We have deleted content related to cryopreservation.

Point 17: style must be revised: it looks like a copy and paste of a bench protocol (from line 169 on).

Response 17: Line 198-201 We have modified the style of this section.

Point 18: English must be revised: the SEs needed to be partial desiccated before germination; mLV medium supplemented WITH 2 g ….; embryos were PLACED and not inoculated onto germination medium.

Response 18: Line 168-177 We revised the English grammar in the article to make the article better understand.

Point 19: I also suggest to replace the world “inoculated” with the correct word “placed” throughout all the text.

Response 19: Line 173 ‘inoculated’ has been replaced by ‘placed’

Point 20:I cannot understand if fig. 1 refers to 2015 or 2017

Response 20: Line 212 We have added the collection time of the material in Fig. 1

Point 21: at line 258 You stated “24 families”, but in the introduction You stated 27. Which data is correct?

Response 21: The 27 families in the introduction refer to the sum of 3 families collected in 2015 and 24 families collected in 2017, and line 258 discusses the results of the embryogenic callus induction rate of 24 families in 2017.

Point 22: Line282: “the proliferation efficiency of embryogenic cell line was general”: please clarify this sentence

Response 22: Line 289-290 ‘the proliferation efficiency of embryogenic cell line was general’ has been replaced by ‘the proliferation efficiency of embryogenic cell line was less than combination of 9.04 μmol·L-1 2,4-D and 4.4 μmol·L-1 6-BA’

Point 23: Paragraph 3.2.2. Each time You state “was less” or “was more” please make a comparison: less than…; more than…

Response 23: Line 307,312We have revised the content of this part.

Point 24: Paragraph 3.3. provide evidences or delete this paragraph.

Response 24: We have removed Paragraph 3.3.

Point 25: Pictures/tables: pictures and tables are nice and clearly describes the results, however there are some typos in the captions, please check them.

Response 25: We have made changes.

Point 26: 4.1 Please discuss the improvement/difference with your previous work in Korean pine (Gao et al 2020) and provide reference at row 345.

Response 26: We have added a discussion of the differences and improvements between this article and previous research results.

Point 27: 4.2 You can incorporate one-two sentences in the previous paragraph.

Response 27: We have deleted the cryogenic preservation part.

Point 28: 4.3 line 389: change the sentence with “When SE were partially desiccated, highest germination percentage was 66%”

Response 28: Line 386 ‘In this study, The SEs Were partial desiccation treatment, highest gemination percentage were 66%.’ has been replaced by ‘When SE was partially desiccated, the highest germination percentage was 66%.’

Point 29: As a general suggestion, punctuation must be revised (many sentences started with “and”, commas and semi-colon needs an extensive English review) and English extensively revised. Further, paragraph in math and methods and results should follow the same order.

Response 29: We have revised the English grammar in the full text to make the article easier to understand.

Round 2

Reviewer 2 Report

Dear authors,

This is an interesting manuscript, filling the gap in the previous methods on Korean pine regeneration. Although the interest in this plant could be geographically limited, the importance of biodiversity preservation is of great importance for a general audience.

I really appreciate the efforts You have done in ameliorating the manuscript; now the writing is clear and all the experiments are well defined and described in materials and methods. Regarding the cryopreservation protocol, I would suggest to provide more evidence and write a short note for one of the many journals specifically focussing in this topic.